# VISE: Variational Integration with Symbolic Expressions

## Abstract

We introduce VISE, a novel family of numerical integration methods that combines symbolic regression with structure-preserving variational integrators to efficiently solve Lagrangian and Hamiltonian ordinary differential equations. Unlike general symbolic regression methods, which seek a single static closed-form expression that may diverge outside the training data or fail to form a valid expression, VISE operates as a hybrid numerical integrator that combines the interpretability of symbolic expressions with the structure-preservation of variational methods. The symbolic expressions serve as adaptive approximation functions with fixed forms whose parameters are dynamically updated through discrete Euler-Lagrange equations at each time step. VISE maintains interpretability and inherits the symplectic structure of Hamiltonian systems. Compared to conventional numerical integrators, VISE allows for larger time steps and fewer degrees of freedom, making it potentially suitable for high-dimensional, time-dependent problems where traditional solvers are computationally prohibitive. By leveraging the strengths of both symbolic regression and variational integrators, VISE offers a promising approach for efficiently and accurately solving complex time-dependent systems.

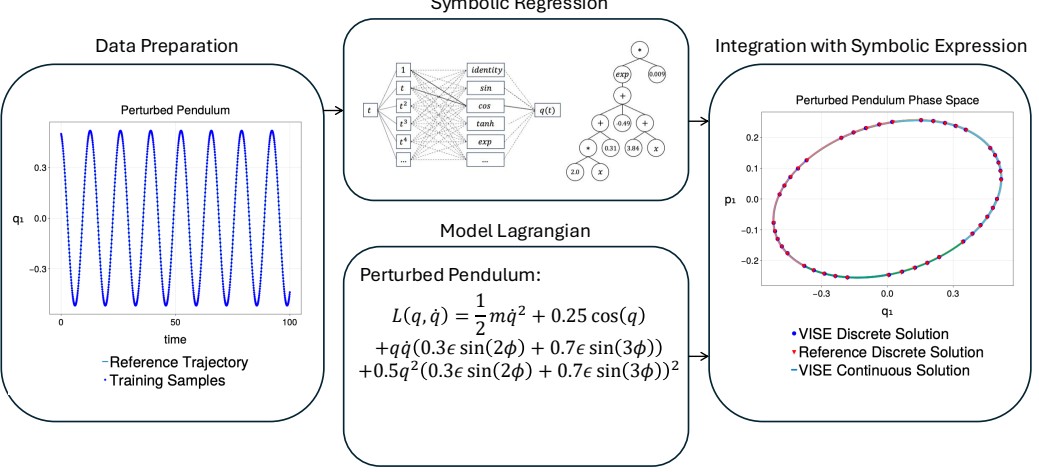

Figure 1: Given observation points from generalized trajectory, an initial parameterized symbolic expression is obtained by symbolic regression methods. Parameters are solved and updated in each time step by discrete Euler-Lagrange equations with given model Lagrangian for every integration step. Continuous solution approximation between discrete time steps are obtained by evaluating the resulting expressions.

# 1 INTRODUCTION

The numerical solution of time-dependent differential equations, particularly Hamiltonian (Leimkuhler & Reich, 2009; Bridges & Reich, 2006) and Lagrangian systems (Lew & Mata A, 2016; Marsden & West, 2001), presents a fundamental challenge in computational physics and engineering. While traditional numerical integrators provide reliable solutions, they often require fine temporal discretization and can be prohibitively expensive to solve for high-dimensional systems due to the curse of dimensionality. Structure-preserving integrators (Vujanovic & Atanackovic, 2004; Hairer et al., 2006), such as variational and symplectic methods, address some of these limitations by maintaining geometric properties like energy conservation. However, the computational cost of high order methods remains significant.

Recently, neural networks and machine learning methods have been applied to solve differential equations. In particular, physics-informed machine learning and related approaches (Raissi et al., 2019; E & Yu, 2017) have demonstrated the potential of neural networks for numerically solving PDEs. Complementary to these methods, Neural ODE (Chen et al., 2019) recasts neural networks as continuous-time dynamical systems governed by ODEs. This perspective not only provides new tools for modelling time-dependent processes but also deepens the connection between ODE theory and deep learning. Specifically, in the context of Hamiltonian and Lagrangian systems, networks are trained to learn a unknown functional, such as the Hamiltonian (Greydanus et al., 2019; Bertalan et al., 2019; Zhong et al., 2019; Toth et al., 2019), Lagrangian (Cranmer et al., 2020a), inverse modified Lagrangian (Ober-Blöbaum & Offen, 2023), and discrete Lagrangian (Offen & Ober-Blöbaum, 2023; 2024; Lishkova et al., 2022), and serve either as surrogate models or as operators between states (Raonić et al., 2023; Jin et al., 2020; Chen et al., 2020; Chen & Tao, 2021; Tong et al., 2021) based on discrete observations. Some of these methods include the structure in the architecture of the neural networks, and are therefore referred to as structure-preserving deep learning methods. Rather than numerically solving the differential equations derived from a known Lagrangian or Hamiltonian, these trained surrogate models learn the system's evolution directly from data, enabling faster predictions but limited generalization. These methods are often constrained by the heavy burden of training and the requirement for large amounts of data, which may be difficult to obtain in practice. Moreover, the surrogate models are typically tied to the resolution of the training data, restricting their ability to evaluate the system beyond discrete time steps. Even when endowed with physical meaning, such neural networks largely retain their black-box nature, offering limited interpretability into the learned parameters.

In contrast, symbolic expressions and symbolic regression methods offer an alternative parameterization, by searching for closed-form analytical governing equations directly from data (Champion et al., 2019; Cranmer et al., 2020b; Tenachi et al., 2023; Cornforth & Lipson, 2012; Brunton et al., 2015; Fiorini et al., 2025; Cranmer, 2023; Meidani et al., 2024; Quade et al., 2016; Lample & Charton, 2019; Khan, 2023). These methods typically involve searching through a library of candidate functions and combining them to find a parsimonious expression that fits the data and often is compact and interpretable. The aforementioned works often focus on recovering the governing vector fields or variational structures such as the Hamiltonian or Lagrangian, while the evolution of the system and the preservation of invariants are typically ensured by incorporating geometric integrators (Dipietro et al.). However, even with access to the vector field, Hamiltonian, or Lagrangian, it remains difficult to obtain long-time scale descriptions of the system in terms of its solutions or generalized trajectories. Besides, most dynamics may not be captured by a single governing equation, and the resulting expressions may fit the training data but can potentially diverge outside the provided scope or even fail to form a valid expression. Thus, attempts to approximate generalized trajectories with symbolic expressions often suffer from non-uniqueness and poor preservation of energy (See Appendix E for comparisons on energy evolution of single global symbolic expressions and VISE), limiting their effectiveness as faithful system representations.

Motivated by these issues, we present a new method named **VISE**, which combines **V**ariational **I**ntegration with **S**ymbolic **E**xpressions. The proposed method enables interpretable, structure-informed modeling of Hamiltonian dynamics. Instead of seeking a single global expression with fixed parameters that satisfies the differential equations, VISE constructs symbolic expressions over local time intervals. In order to evolve the system, parameters in the initial symbolic expression are updated at each step by solving discrete Euler–Lagrange equations derived from variational principles. In this way, VISE behaves more like a traditional numerical integrator rather than a purely

data-driven surrogate model, and it is not restricted to the training region or step size of training data. After the parameters in symbolic expression are solved in each time step, the solution can be evaluated continuously within the interval, so the solution for the whole time interval is a piece-wise continuous symbolic function instead of one global expression. Moreover, the method preserves the system's energy (Appendix D), while simultaneously enabling model reduction: fewer degrees of freedom are sufficient to integrate the system with larger time steps, where inherent symmetry and periodicity can also be observed.

## 2 THEORY

**Lagrangian and Hamiltonian ODEs, Variational Principle.** Classical mechanics has two main points of view, the Lagrangian mechanics and Hamiltonian mechanics. Here we briefly review Lagrangian and Hamiltonian mechanics. For more comprehensive description, we refer to Igorevich (1978); Marsden & Wendlandt (1997); Marsden & Ratiu (1999).

In Lagrangian mechanics, the equation of motion for $d$-dimension system has the form:

$$\frac{\partial \mathcal{L}}{\partial q}\big(q(t), \dot{q}(t)\big) - \frac{d}{dt}\frac{\partial \mathcal{L}}{\partial \dot{q}}\big(q(t), \dot{q}(t)\big) = 0, \tag{1}$$

where $\mathcal{L}$ is the Lagrangian functional, $q = \big(q^1, ..., q^d\big) \in R^d$ is the generalized trajectory, and $\dot{q}$ is the generalized velocity. Introducing the generalized momentum $p$, the second order differential equation of motion can be rewritten as a system of first order differential equations, the Hamiltonian equations:

$$\dot{q}(t) = \frac{\partial \mathcal{H}}{\partial p}\big(q(t), p(t)\big), \quad \dot{p}(t) = -\frac{\partial \mathcal{H}}{\partial q}\big(q(t), p(t)\big), \tag{2}$$

or in a more compact form:

$$J z_t = \nabla_z \mathcal{H}(z), \quad \text{with } z = \begin{pmatrix} q \\ p \end{pmatrix}, \quad J = \begin{pmatrix} 0 & I_d \\ -I_d & 0 \end{pmatrix}, \tag{3}$$

where the Hamiltonian functional $\mathcal{H}$ is defined as the Legendre transform of the Lagrangian, often interpreted as the total energy of the system. $J$ is the skew-symmetric symplectic matrix.

The Lagrangian equation of motion could be derived from a variational principle, specifically Hamilton's principle of stationary action, which states that considering all possible trajectories $q$ a system could follow to get from initial state $q(0)$ to end state $q(T)$, the following integral, called the action:

$$\mathcal{A}[q] = \int_0^T \mathcal{L}\big(q(t), \dot{q}(t)\big)dt, \tag{4}$$

is stationary for the actual physical trajectory $q$, i.e. the variation of the action is zero, $\delta \mathcal{A} = 0$. With Hamilton's principle, one can construct variational integrators for the Lagrangian ordinary differential equations. This involves discretizing the time interval $[0, T]$ into $N$ equally spaced time intervals, $t_n = nh$, $n = 0, 1, ..., N$, with $h = T/N$. By defining the exact discrete Lagrangian $\mathcal{L}_d^e(q_n, q_{n+1})$ in each interval, the discrete action has the form:

$$\mathcal{A}[q] = \sum_{n=0}^{N-1} \int_{t=t_n}^{t_{n+1}} \mathcal{L}\big(q(t), \dot{q}(t)\big)dt = \sum_{n=0}^{N-1} \mathcal{L}_d^e\big(q_n, q_{n+1}\big) \approx \sum_{n=0}^{N-1} \mathcal{L}_d\big(q_n, q_{n+1}\big), \tag{5}$$

where $\mathcal{L}_d(q_n, q_{n+1})$ is the discrete Lagrangian, an approximation of the exact discrete Lagrangian. By taking the variation of the discrete action with respect to each $q_n$ and $q_{n+1}$, and require the discrete trajectory $\{q_n\}_{n=1}^N$ satisfy a discrete version of Hamilton's principle, we obtain the discrete Euler-Lagrange equations:

$$D_1 \mathcal{L}_d\big(q_n, q_{n+1}\big) + D_2 \mathcal{L}_d\big(q_{n-1}, q_n\big) = 0, \quad n = 1, ..., N - 1, \tag{6}$$

where $D_i$ denotes the derivative with respect to the $i$th argument. The discrete Euler-Lagrange equations could be seen as a one-step integrator for the Lagrangian ordinary differential equations, while one could also rewrite into a position-momentum form:

$$p_n = -D_1 \mathcal{L}_d\big(q_n, q_{n+1}\big), \quad p_{n+1} = D_2 \mathcal{L}_d\big(q_{n-1}, q_n\big). \tag{7}$$

A variant of the above variational principle is the Hamilton-Pontryagin principle (Bou-Rabee & Marsden, 2009; Yoshimura & Marsden, 2006a;b):

$$\mathcal{A}[q, v, p] = \int_0^T L\big(q(t), v(t)\big)dt + \int_0^T \langle p(t), \dot{q}(t) - v(t)\rangle dt, \tag{8}$$

which in addition to the plain action, a kinematic constraint relates the mechanical system's velocity to a curve on the tangent bundle is imposed via a Lagrange multiplier $p$, the generalized momentum.

**Galerkin Variational Integrators.** By discretizing the action integral directly, one can obtain a variational integrator, which is automatically symplectic (Marsden & Wendlandt, 1997; Wendlandt & Marsden, 1997). Though they do not preserve the Hamiltonian exactly, variational integrators can still provide good long-term behavior for many physical systems, as the derivation of numerical energy from the true energy is bounded over exponentially long time (Hairer et al., 2006; Hairer & Lubich, 1997). We briefly review the Galerkin approach for constructing a high-order variational integrator (Campos, 2014; Ober-Blöbaum & Saake, 2014; Ober-Blöbaum, 2016). For simplicity, we consider the case of a one-dimensional system.

The approximation of the action integral consists of two steps: Approximate the infinite dimensional space of the generalized trajectory $q$ with a finite dimensional subspace, apply a quadrature rule for the discrete action integral.

Given a set of basis functions $\{\phi_j(x)\}_{j=1}^S$, e.g. Lagrangian polynomials, defined on $[0, 1]$, the generalized trajectory $q$ and generalized velocity $\dot{q}$ can be approximated as

$$q_h^n(t) = \sum_{j=1}^S Q_{n,j}\phi_j(\frac{t - t_n}{h}), \quad \dot{q}_h^n(t) = \frac{1}{h}\sum_{j=1}^S Q_{n,j}\frac{d}{dx}\phi_j(\frac{t - t_n}{h}), \quad t \in [t_n, t_{n+1}], \tag{9}$$

where $Q_{n,j}$ are the coefficients to be determined. The continuity constraint of generalized trajectory $q$ between time steps are enforced by the condition $q_h^n(t = t_{n+1}) = q_h^{n+1}(t = t_{n+1})$ implicitly. Here we write this constraint explicitly similar to Hamilton-Pontryagin principle in the discretization of action integral, but restrict the discrete generalized trajectory instead of generalized velocity:

$$\mathcal{A} = \sum_{n=0}^{N-1}\left\{\int_{t_n}^{t_{n+1}}\mathcal{L}\big(q_h^n(t), \dot{q}_h^n(t)\big)dt + p_n\big[q_h^n(t_n) - q_n\big] + \bar{p}_{n+1}\big[q_{n+1} - q_h^n(t_{n+1})\big]\right\}$$

$$\approx \sum_{n=0}^{N-1}\left\{h\sum_{r=1}^R \alpha_r\mathcal{L}_d\big(\sum_{j=1}^S Q_{n,j}\phi_j(\beta_r), \frac{1}{h}\sum_{j=1}^S Q_{n,j}\dot{\phi}_j(\beta_r)\big)\right. \tag{10}$$

$$\left. + p_n\big[\sum_{j=1}^S Q_{n,j}\phi_j(0) - q_n\big] + \bar{p}_{n+1}\big[q_{n+1} - \sum_{j=1}^S Q_{n,j}\phi_j(1)\big]\right\},$$

where $R$ is the number of quadrature points, $\alpha_r$ are the quadrature weights, and $\beta_r$ are the quadrature nodes. In the comparison of VISE with Galerkin variational integrator, we use Gauss-Legendre quadrature points as the nodes of Lagrangian polynomial basis functions $\phi_j(x)$, i.e., $R = S$ for implementation. A comprehensive derivation on discretizing the action can be found in Appendix B.

## 3 VARIATIONAL INTEGRATION WITH SYMBOLIC EXPRESSIONS

In the construction of Galerkin variational integrators, the basis functions $\phi_j(x)$ are typically chosen as polynomials defined on the reference interval $[0, 1]$, and then scaled to each time interval $[t_n, t_{n+1}]$. In certain mechanical systems, the generalized trajectory $q$ exhibits specific structures such as periodicity, symmetry, or some prior knowledge about the system's behavior is available. In these cases, incorporating these properties into the simulation and integration may lead to more accurate and efficient numerical methods, as opposed to relying on generic polynomial bases.

In this section, we propose to use symbolic expressions to approximate the generalized trajectory $q$ and insert the expressions into the action functional, by which the discrete Euler-Lagrange equations of the parameters in symbolic expressions can be derived.

**Discretization with Symbolic Expressions.** The action functional in this case is quite straightforward and similar to Galerkin variational integrators. Suppose we have a symbolic expression $q(t; \theta_n)$ to approximate the trajectory between $[t_n, t_{n+1}]$, where $\theta_n = \{\theta_{n,j}\}_{j=1}^S$ denotes parameters. The approximating generalized velocity $\dot{q}(t; \theta_n)$ can be computed as the derivative of $q(t; \theta_n)$ with respect to time analytically. The action can be written as:

$$
\mathcal{A} \approx \sum_{n=0}^{N-1} \left\{ \int_{t_n}^{t_{n+1}} \mathcal{L}\big(q(t; \theta_n), \dot{q}(t; \theta_n)\big) dt + p_n\big[q(t; \theta_n) - q_n\big] + \bar{p}_{n+1}\big[q_{n+1} - q(t; \theta_n)\big] \right\}
$$

$$
\approx \sum_{n=0}^{N-1} \left\{ h \sum_{r=1}^{R} \alpha_r \mathcal{L}_d\big(q(t; \theta_n), \dot{q}(t; \theta_n)\big) \right. \tag{11}
$$

$$
\left. + p_n\big[q(t; \theta_n) - q_n\big] + \bar{p}_{n+1}\big[q_{n+1} - q(t; \theta_n)\big] \right\}.
$$

Computing the variation of the action functional with respect to all parameters $\theta_n$, $p_n$, and $\bar{p}_{n+1}$ leads to the discrete Euler-Lagrange equations to integrate the system.

$$
\delta\mathcal{A}(\theta_n, q_n, q_{n+1}, p_n, \bar{p}_{n+1})
$$

$$
= \sum_{n=0}^{N-1} \sum_{j=1}^{S} \left[ h \sum_{r=1}^{R} \alpha_r \frac{\partial \mathcal{L}}{\partial q} \frac{\partial q}{\partial \theta_{n,j}}\big|_{t=t_n+h\beta_r} + \sum_{r=1}^{R} \alpha_r \frac{\partial \mathcal{L}}{\partial \dot{q}} \frac{\partial \dot{q}}{\partial \theta_{n,j}}\big|_{t=t_n+h\beta_r} \right.
$$

$$
\left. + p_n \frac{\partial q}{\partial \theta_{n,j}}\big|_{t=t_n} - \bar{p}_{n+1} \frac{\partial q}{\partial \theta_{n,j}}\big|_{t=t_{n+1}} \right] \delta\theta_{n,j} \tag{12}
$$

$$
+ \sum_{n=0}^{N-1} \left[ q(t_n; \theta_n) - q_n \right] \delta p_n + \sum_{n=0}^{N-1} \left[ q_{n+1} - q(t_{n+1}; \theta_n) \right] \delta\bar{p}_{n+1}
$$

$$
+ \sum_{n=0}^{N-1} (-p_n)\delta q_n + \sum_{n=0}^{N-1} \bar{p}_{n+1}\delta q_{n+1},
$$

where $R$ is the number of quadrature points, $\alpha_r$ are the quadrature weights, and $\beta_r$ are the quadrature nodes. The derivative of $q$ and $\dot{q}$ with respect to parameters $\theta_n$ can be computed analytically. Following the discrete version of Hamilton action principle, i.e., the variation of all variables should vanish, the discrete Euler-Lagrange equations for VISE are:

$$
0 = h \sum_{r=1}^{R} \alpha_r \frac{\partial \mathcal{L}}{\partial q} \frac{\partial q}{\partial \theta_{n,j}}\big|_{t=t_n+h\beta_r} + \sum_{r=1}^{R} \alpha_r \frac{\partial \mathcal{L}}{\partial \dot{q}} \frac{\partial \dot{q}}{\partial \theta_{n,j}}\big|_{t=t_n+h\beta_r}
$$

$$
+ p_n \frac{\partial q}{\partial \theta_{n,j}}\big|_{t=t_n} - \bar{p}_{n+1} \frac{\partial q}{\partial \theta_{n,j}}\big|_{t=t_{n+1}}, \quad j = 1, 2, ..., S, n = 0, 1, ..., N-1, \tag{13a}
$$

$$
0 = q(t_n; \theta_n) - q_n, \quad n = 0, 1, ..., N-1, \tag{13b}
$$

$$
0 = q_{n+1} - q(t_{n+1}; \theta_n), \quad n = 0, 1, ..., N-1, \tag{13c}
$$

$$
0 = p_n - \bar{p}_n, \quad n = 1, 2, ...N-1. \tag{13d}
$$

In every time step, $S + 1$ equations (13a–13b) are solved for $S + 1$ unknowns $\{\theta_{n,j}\}_{j=1}^S$ and $p_{n+1}$ with nonlinear solvers. After solving the parameters $\theta_n$, the generalized trajectory between $t \in [t_n, t_{n+1}]$ can be evaluated by the symbolic expression $q(t; \theta_n)$.

**Pretraining the Symbolic Expressions.** In order to obtain a symbolic expression for generalized trajectory $q$, we employ the analytic solution of the differential equation, if available, or high fidelity integrators to generate training data. Then we use models proposed in Fiorini et al. (2025) with the root mean squared error and Cranmer (2023) to perform symbolic regression. The symbolic expressions may vary depending on the training configurations. Here, we manually choose one of the resulting expressions with a good balance between accuracy and complexity as the initial expression to demonstrate the effectiveness of the proposed method. The same symbolic expression is then applied for different time steps at the beginning of the integration. The initial conditions and amount of training data to obtain the symbolic expressions are summarized in Table 1 in Appendix C.

## 4 NUMERICAL EXPERIMENTS

We evaluate VISE on three conservative Hamiltonian systems, comparing it with the implicit midpoint integrator and Galerkin variational integrators. Our evaluation focuses on: (1) accuracy vs. computational cost trade-offs, (2) long-term stability, and (3) energy preservation. All methods are tested with time steps $h \in \{1.0, 2.0, 5.0\}$ over extended integration periods longer than the training interval. We utilize Newton's method with backtracking line search to solve the discrete Euler-Lagrange equations for VISE, implicit midpoint integrator, and Galerkin variational integrator. For VISE, the initial guess for the nonlinear solver is the solution of the previous time step, while the reference integrators use Hermite extrapolation from the solutions of the previous two time steps. For the first integration step, the parameters from symbolic regression are used as the initial guess of nonlinear solver.

The number of quadrature points $R$ generally increases with larger step sizes across all examples. Additionally, the number of basis functions for Galerkin variational integrators, $S = R$, is always larger than the number of parameters in the symbolic expressions. We also compare and record the relative errors in generalized trajectory and Hamiltonian conservation when the degrees of freedom are equal for VISE and Galerkin variational integrators. A detailed table corresponding to the figures below and extra comparisons can be found in Appendix D.

**Harmonic Oscillator.** The first example is the harmonic oscillator, which is a simple mechanical system with a well-known analytical solution and shows rich geometric structures. The Lagrangian of the harmonic oscillator is given by:

$$\mathcal{L}(q, \dot{q}) = \frac{1}{2}\dot{q}^2 - \frac{1}{2}kq^2, \tag{14}$$

where $k$ is the spring constant. We set $k = 0.5$ for simplicity. The system has the analytical solution:

$$q(t) = A\cos(\sqrt{k}t + \phi), \text{with } A = \sqrt{q_0^2 + \frac{p_0^2}{k}}, \quad \phi = -\arctan\left(\frac{p_0}{\sqrt{k}q_0}\right). \tag{15}$$

In this example, the initial symbolic expression shares the same structure as the analytical solution and VISE's solution shows better accuracy than that of Galerkin variational integrators with fewer degrees of freedom, as shown in Figure 2. As the implicit midpoint is unconditionally stable for linear harmonic oscillators, it can preserve the energy exactly, but a phase shift is observed in the generalized trajectory when we increase the step size.

**Perturbed Pendulum.** The Lagrangian of the perturbed pendulum is given by:

$$\mathcal{L}(q, \dot{q}) = \frac{1}{2}m\dot{q}^2 + \omega^2 \cos(q)$$
$$+ q\dot{q}\left(0.3\epsilon\sin(2\phi) + 0.7\epsilon\sin(3\phi)\right) + 0.5q^2\left(0.3\epsilon\sin(2\phi) + 0.7\epsilon\sin(3\phi)\right)^2, \tag{16}$$

where $m$ is the mass of the pendulum, $\omega$ is the angular frequency, and $\epsilon$ is a small parameter that controls the strength of the perturbation. We set $m = 1$ and $\omega = 0.5$, $\epsilon = 0.5$ and $\phi = \frac{\pi}{3}$ when generating the training data. As we can observe in Figure 3, VISE provides a reasonable approximation of the generalized trajectory and Hamiltonian conservation, even though the symbolic expression is not the exact solution of the system. Particularly, VISE outperforms the implicit midpoint method with large step sizes.

**Hénon-Heiles Potential.** The third example is the Hénon-Heiles Potential model, which serves as a simple example of a system that exhibits chaotic behavior. The Lagrangian of the Hénon-Heiles Potential model is given by:

$$\mathcal{L}(q, \dot{q}) = \frac{1}{2}\dot{q}^2 - \frac{1}{2}q^2 - \lambda\left(q_1^2 q_2 - \frac{q_2^3}{3}\right), \quad \text{with } q = (q_1, q_2)^T, \tag{17}$$

where $\lambda$ is a coupling constant, and we set $\lambda = 1$. When the initial energy is lower than the critical escape energy ($E_c = \frac{1}{6}$ if $\lambda = 1$), the system exhibits quasi-periodic behavior and the generalized trajectory and generalized momentum are often bounded and ordered in phase space. The amplitudes for both generalized trajectory and momentum oscillations are not constant, but change slowly over time with the given initial conditions. A global symbolic expression for the generalized trajectory is

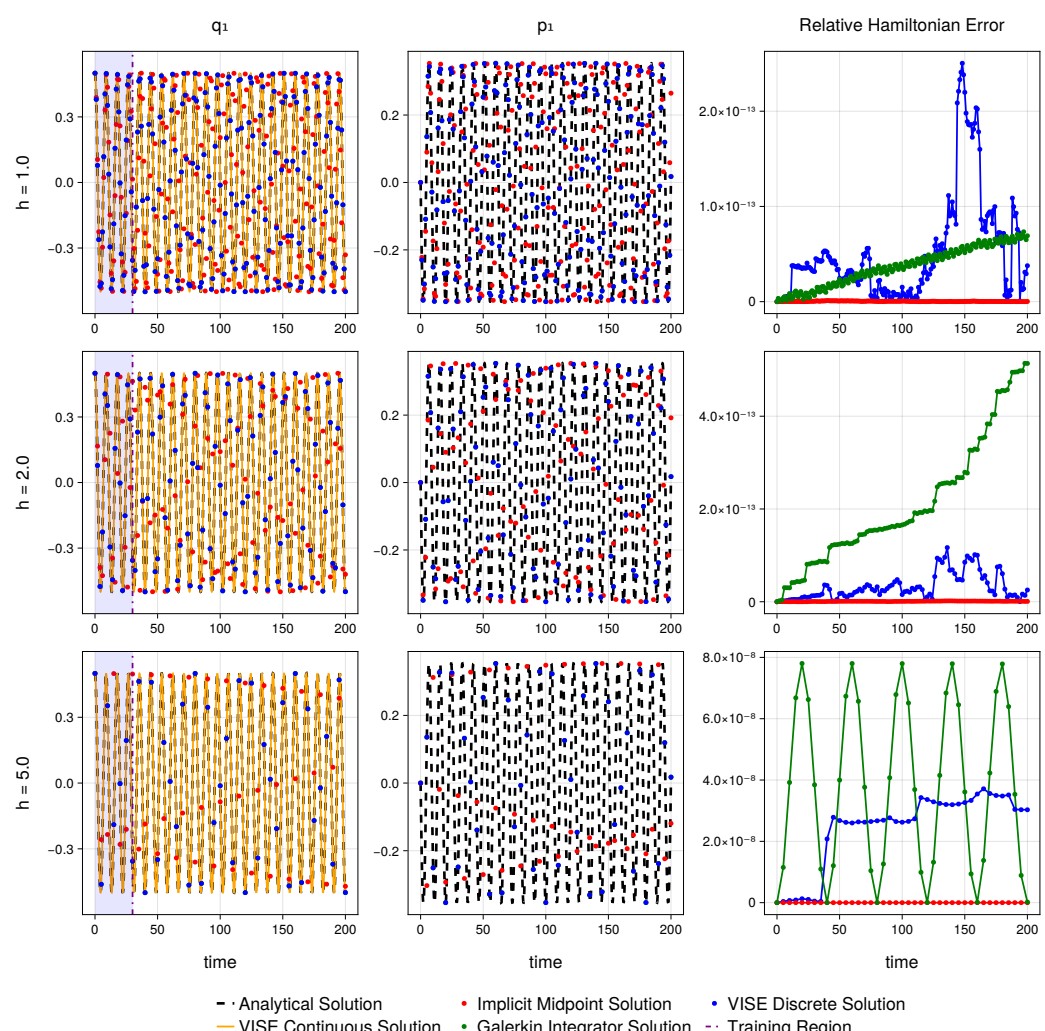

Figure 2: **Harmonic Oscillator.** With different time step sizes $h = 1.0, 2.0$ and $5.0$, implicit midpoint integrator, Galerkin variational integrator, and VISE are compared. The training data are generated before the dash dotted vertical line $t = 30.0$. Though the implicit midpoint method preserves the Hamiltonian exactly, a phase shift is observed in the generalized trajectory when we increase the step size, while VISE does not show this behavior. VISE's solution shows better accuracy than that of Galerkin variational integrators with fewer degrees of freedom, while the energy error is also smaller and bounded.

not available in this case and VISE can provide a piecewise continuous symbolic expression for the generalized trajectory instead.

The results for Hénon-Heiles Potential are shown in Figure 4. VISE results are comparable to implicit midpoint for $h = 1.0, 2.0$, while the implicit midpoint cannot provide a reasonable approximation of the generalized trajectory with large step sizes, $h = 5.0$.

## 5 DISCUSSION AND CONCLUSION

We have proposed VISE (Variational Integrator with Symbolic Expressions), a novel numerical integration scheme that unifies the strengths of variational integrators with the interpretability of symbolic regression. The key idea is to represent generalized trajectories using parsimonious symbolic expressions, which are then inserted into the action functional. At each time step, the parameters

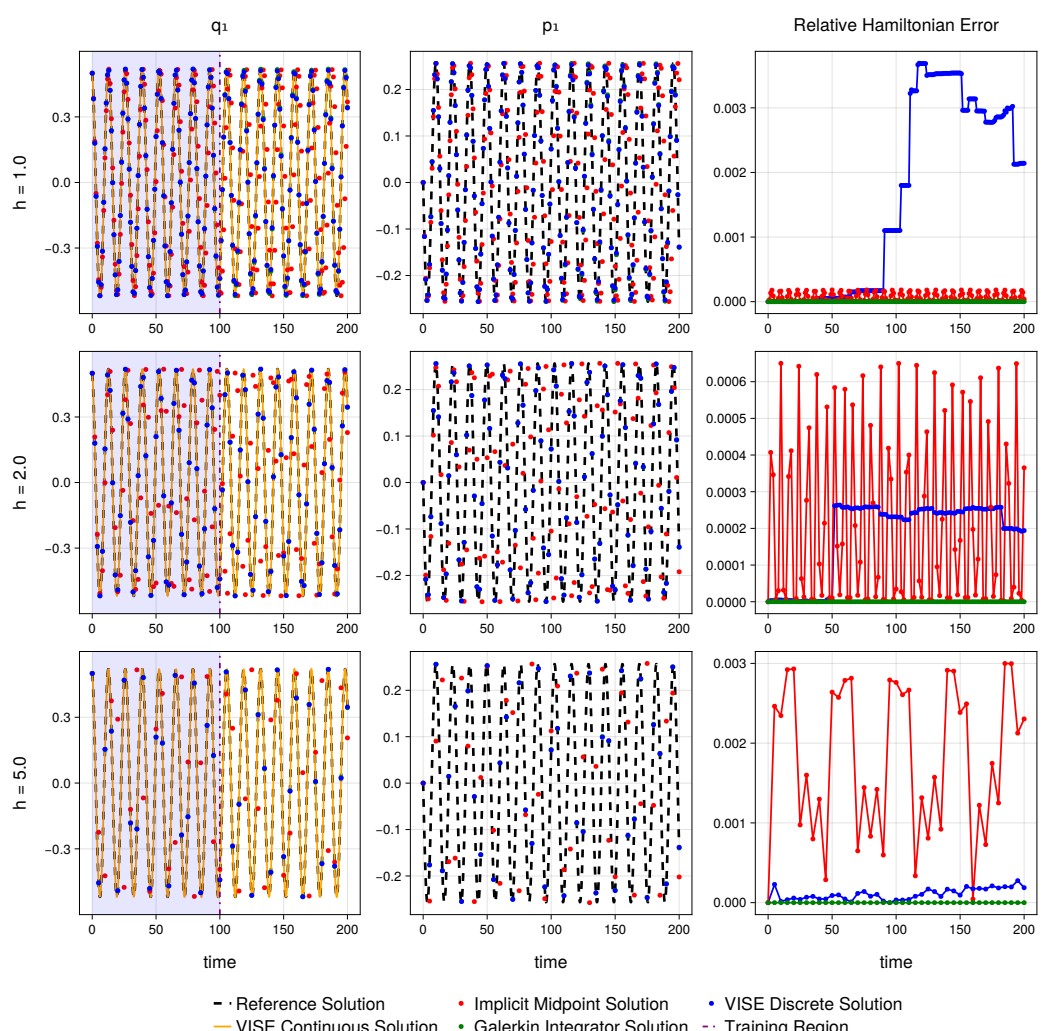

Figure 3: **Perturbed Pendulum**. As this problem does not have an analytical solution, the symbolic expressions are just parsimonious approximations. When $h = 1.0$, the VISE solution exhibits a number of sharp changes in the relative Hamiltonian error that can be attributed to convergence issues of the nonlinear solver. When $h = 2.0, 5.0$, the oscillations in Hamiltonian error of VISE are smaller than those of the implicit midpoint, indicating better energy conservation.

of these symbolic expressions are constrained by the Euler-Lagrange equations and updated accordingly, yielding a piece-wise continuous solution for the underlying Lagrangian system.

This construction preserves the geometric structure of the dynamics while reducing the computational cost of simulation. Unlike most machine learning-based surrogates, VISE leverages symbolic regression that explicitly contains system structure, enabling both interpretability and model reduction. By adapting the symbolic expressions sequentially, VISE avoids the drift in accuracy and instability often observed in fixed-parameter symbolic surrogates.

Our numerical experiments (Appendix D) demonstrate that VISE significantly outperforms Galerkin variational integrators when both methods are given the same number of degrees of freedom. In particular, VISE retains high accuracy and stable energy conservation even for large time step sizes, where standard integrators typically deteriorate. This makes VISE suited for problems where long-term stability and conservation laws are critical.

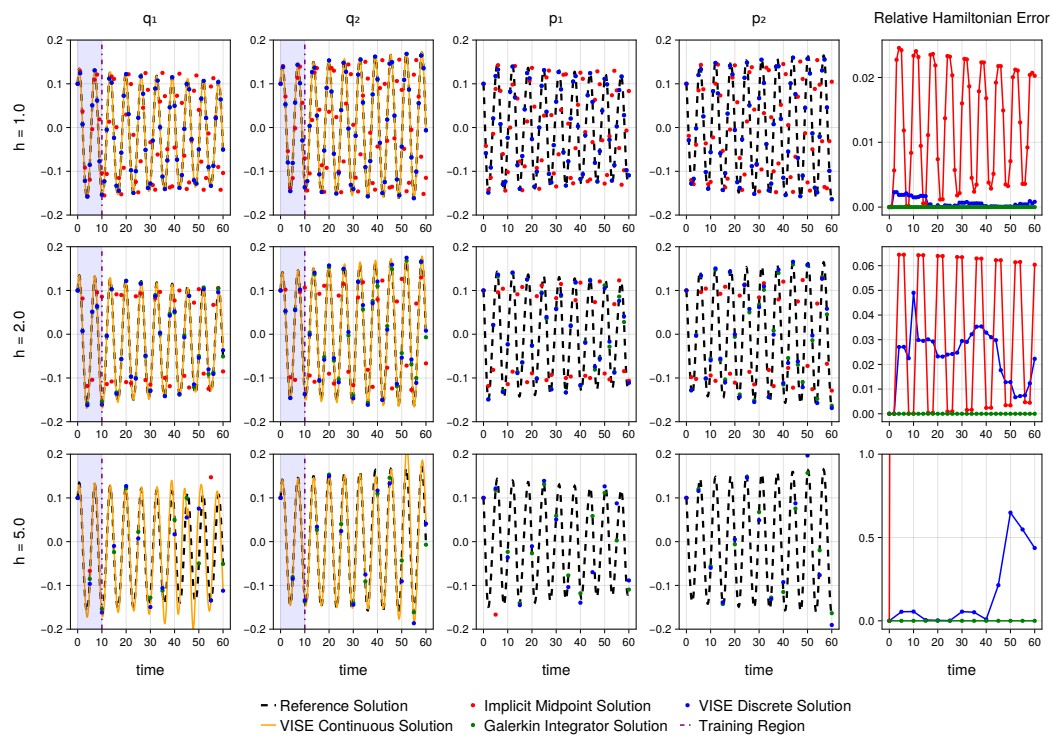

Figure 4: **Hénon-Heiles Potential.** The magnitude of the generalized trajectory and momentum are not constant, but changing slowly over time, different from the previous two examples. VISE is suitable and is able to capture this behavior effectively considering the pattern of generalized trajectory.

From a broader perspective, VISE can be interpreted as an adaptive symbolic model reduction framework. Because it requires fewer degrees of freedom to evolve the system, it reduces computational complexity while retaining accuracy. Moreover, domain knowledge can be naturally encoded into the symbolic expressions, allowing the method to leverage known invariants or symmetries in the system. This is a key advantage over both black-box deep learning surrogates, which sacrifice interpretability, and purely numerical schemes, which may fail to exploit available structural information.

While the present work provides a proof of concept, several important research directions remain. These include developing principled metrics for assessing the fitness of symbolic expressions to a given system, improving strategies for initializing the nonlinear solvers, and extending VISE to partial differential equations and stochastic systems. We anticipate that these extensions, coupled with advances in symbolic regression, will further enhance the applicability of VISE to high-dimensional and multi-scale dynamical systems.

In summary, VISE provides a promising new paradigm for geometric integration, combining the rigor of variational methods with the flexibility and interpretability of symbolic modeling. By offering both energy stability and parsimonious representations, it opens the door to efficient, explainable, and structure-preserving simulation across a wide range of scientific and engineering domains.

## 6 REPRODUCIBILITY STATEMENT

Given the initial conditions and parameters in Table 1 in Appendix C and the given discrete Euler-Lagrange equations, all the results in this paper can be reproduced with standard numerical libraries.

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

# A  USAGE OF LARGE LANGUAGE MODEL

During the process of writing this paper, we used GPT-4 and Google-AI studio to help polish the language and grammar of the manuscript. All ideas, experiments, and analyses were conducted by the authors.

# B  GALERKIN VARIATIONAL INTEGRATORS

With the proposed action functional,

$$
\begin{aligned}
\mathcal{A} &= \sum_{n=0}^{N-1}\left\{\int_{t_n}^{t_{n+1}}\mathcal{L}\big(q_h^n(t),\dot{q}_h^n(t)\big)dt + p_n\big[q_h^n(t_n)-q_n\big]+\bar{p}_{n+1}\big[q_{n+1}-q_h^n(t_{n+1})\big]\right\} \\
&\approx \sum_{n=0}^{N-1}\left\{h\sum_{r=1}^{R}\alpha_r\mathcal{L}_d\big(\sum_{j=1}^{S}Q_{n,j}\phi_j(\beta_r),\frac{1}{h}\sum_{j=1}^{S}Q_{n,j}\dot{\phi}_j(\beta_r)\big)\right. \\
&\quad \left.+p_n\big[\sum_{j=1}^{S}Q_{n,j}\phi_j(0)-q_n\big]+\bar{p}_{n+1}\big[q_{n+1}-\sum_{j=1}^{S}Q_{n,j}\phi_j(1)\big]\right\},
\end{aligned}
\tag{18}
$$

where $R$ is the number of quadrature points, $\alpha_r$ are the quadrature weights, and $\beta_r$ are the quadrature nodes. We compute the variation on all degrees of freedom: $Q_n = \{Q_{n,j}\}_{j=1}^{S}$, $q_n$, $q_{n+1}$, $p_n$, $\bar{p}_{n+1}$ at all discrete time $t_n, n = 0, 1, ..., N-1$:

$$
\begin{aligned}
&\delta\mathcal{A}(Q_n,q_n,q_{n+1},p_n,\bar{p}_{n+1}) \\
&= \sum_{n=0}^{N-1}\sum_{j=1}^{S}\left[h\sum_{r=1}^{R}\alpha_r\frac{\partial\mathcal{L}}{\partial q}\big|_{t=t_n+h\beta_r}\phi_j(\beta_r)+\sum_{r=1}^{R}\alpha_r\frac{\partial\mathcal{L}}{\partial\dot{q}}\big|_{t=t_n+h\beta_r}\dot{\phi}_j(\beta_r)\right. \\
&\quad \left.+p_n\phi_j(0)-\bar{p}_{n+1}\phi_j(1)\right]\delta Q_{n,j} \\
&\quad +\sum_{n=0}^{N-1}\left[\sum_{j=1}^{S}Q_{n,j}\phi_j(0)-q_n\right]\delta p_n+\sum_{n=0}^{N-1}\left[q_{n+1}-\sum_{j=1}^{S}Q_{n,j}\phi_j(1)\right]\delta\bar{p}_{n+1} \\
&\quad +\sum_{n=0}^{N-1}(-p_n)\delta q_n+\sum_{n=0}^{N-1}\bar{p}_{n+1}\delta q_{n+1}.
\end{aligned}
\tag{19}
$$

Since all degrees of freedom are required to satisfy the discrete version of Hamilton action principle, i.e., the variation of all variables should vanish, the expressions in square brackets need to be zero, which leads to the discrete Euler-Lagrange equations for Galerkin variational integrators:

$$
0 = h\sum_{r=1}^{R}\alpha_r\frac{\partial\mathcal{L}}{\partial q}\big|_{t=t_n+h\beta_r}\phi_j(\beta_r)+\sum_{r=1}^{R}\alpha_r\frac{\partial\mathcal{L}}{\partial\dot{q}}\big|_{t=t_n+h\beta_r}\dot{\phi}_j(c_j)+p_n\phi_j(0)-\bar{p}_{n+1}\phi_j(1),
$$
$$
j = 1,2,...,S, n = 0,1,...,N-1,
\tag{20a}
$$

$$
0 = \sum_{j=1}^{S}Q_{n,j}\phi_j(0)-q_n, \quad n = 0,1,...,N-1,
\tag{20b}
$$

$$
0 = q_{n+1}-\sum_{j=1}^{S}Q_{n,j}\phi_j(1), \quad n = 0,1,...,N-1,
\tag{20c}
$$

$$
0 = p_n-\bar{p}_n, \quad n = 1,2,...N-1.
\tag{20d}
$$

As suggested in equation (20d), $p_n = \bar{p}_n$. Based on the discrete Legendre transforms and equation (20d), the newly introduced Lagrangian multipliers $p_n$ are generalized momentum. Each integration step needs to solve a $(S+1)\times(S+1)$ system equation (20a,20b) of $(Q_n, p_{n+1})$ iteratively given $(q_n, p_n)$, and use equation (20c) to update $q_{n+1}$.

The purpose of adding two extra constraints is not just to propagate the system explicitly, but also to provide a natural way to enforce the continuity of generalized trajectory $q$ between time steps.

## C    DATA GENERATION FOR TRAINING

To obtain the symbolic expressions for generalized trajectory $q$, we generate training data with high fidelity integrators. An initial symbolic expression is then obtained by performing symbolic regression on the training data. The initial conditions and amount of training data to obtain the symbolic expressions are summarized in Table 1.

Table 1: **Data generation and obtained initial expressions.** By high fidelity integrators, with step size $h$ for duration $t \in [0, T]$, we are able to generate training data. The scalar parameters in obtained initial symbolic expression are then updated in each time step by solving the discrete Euler-Lagrange equations.

Harmonic Oscillator

| | |
|---|---|
| Initial Condition | $q_0 = 0.5, \ p_0 = 0.0$ |
| Sample Domain | $t \in [0.0, 30.0]$ |
| Training Sample Amount N | 300 |
| Initial Symbolic Expression | $q(t) = -0.5000433 \sin(0.70535007t - 1.567814033)$ |
| Degrees of Freedom | 3 |

Perturbed Pendulum

| | |
|---|---|
| Initial Condition | $q_0 = 0.5, \ p_0 = 0.0$ |
| Sample Domain | $t \in [0.0, \ 100.0]$ |
| Training Sample Amount N | 1000 |
| Initial Symbolic Expression | $q(t) = -0.51941 \cos(-0.47405t + 2.8713)$ |
| Degrees of Freedom | 3 |

Hénon-Heiles Potential

| | |
|---|---|
| Initial Condition | $q_0 = (0.1, 0.1), \ p_0 = (0.1, 0.1)$ |
| Sample Domain | $t \in [0.0, \ 10.0]$ |
| Training Sample Amount N | 100 |
| Initial Symbolic Expression | $q_1(t) = 0.14831 \cos(1.0005t - 0.64812) - 0.018712$ 
 $q_2(t) = 0.14298 \cos(-0.97215t + 0.7615) - 0.0013983$ |
| Degrees of Freedom | 8 |

# D    RESULTS OF NUMERICAL EXPERIMENTS

We compare the performance of the proposed VISE with implicit midpoint method and Galerkin variational integrator on each problem, in terms of maximum relative error on generalized trajectory and Hamiltonian at each time step.

$$\text{Relative Error of Generalized Trajectory} : \max_n \left| \frac{q(t_n; \theta) - q_{ref}(t_n)}{q_{ref}(t_n)} \right|,$$

$$\text{Relative Hamiltonian Error} : \max_n \left| \frac{\mathcal{H}(q(t_n; \theta), p(t_n; \theta)) - \mathcal{H}(q_0, p_0)}{\mathcal{H}(q_0, p_0)} \right|, \tag{21}$$

where $q_{ref}(t)$ is the reference solution, $\mathcal{H}(q_0, p_0)$ is the initial Hamiltonian.

The time periods for comparison are $t \in [0, 200]$ for harmonic oscillator and perturbed pendulum, and $t \in [0, 60]$ for Hénon-Heiles potential. We observe that VISE can provide better accuracy than Galerkin variational integrators with the same number of degrees of freedom with large time step sizes. Specifically, VISE achieve errors that are several orders of magnitude smaller than Galerkin variational integrators in both generalized trajectory and Hamiltonian conservation when $h = 2.0, 5.0$, same magnitude when $h = 1.0$ on harmonic oscillator and perturbed pendulum problems. For Hénon-Heiles potential, VISE is comparable to implicit midpoint method when $h = 1.0, 2.0$, while the implicit midpoint method and Galerkin variational integrator with same degrees of freedom cannot provide a reasonable approximation of the generalized trajectory with large step sizes, $h = 5.0$.

Table 2: Numerical results on harmonic oscillator.

| Integrator | h | DOFs | R | Maximum Relative $q$ Error | Maximum Relative Hamiltonian Error |
|---|---|---|---|---|---|
| VISE | 1.0 | 3 | 8 | 6.727209608805657e-12 | 2.503552920529728e-13 |
| Implicit Midpoint | 1.0 | 2 | 1 | 105.11211252647345 | 1.1102230246251565e-15 |
| Galerkin Variational Integrator | 1.0 | 8 | 8 | 8.761490410882109e-13 | 7.338574192772285e-14 |
| | 1.0 | 3 | 3 | 0.5664619043751603 | 0.00036549464750468275 |
| VISE | 2.0 | 3 | 16 | 6.0038777336211566e-12 | 1.1679546219056647e-13 |
| Implicit Midpoint | 2.0 | 2 | 1 | 195.66279738724475 | 1.887379141862766e-15 |
| Galerkin Variational Integrator | 2.0 | 16 | 16 | 3.36738693018362e-12 | 5.140332604014475e-13 |
| | 2.0 | 3 | 3 | 7.084759502833208 | 0.006941287512133076 |
| VISE | 5.0 | 3 | 8 | 6.021457042053391e-7 | 3.7135718056191536e-8 |
| Implicit Midpoint | 5.0 | 2 | 1 | 111.63227353051155 | 5.329070518200751e-15 |
| Galerkin Variational Integrator | 5.0 | 8 | 8 | 1.2362047839725346e-5 | 7.802355472819045e-8 |
| | 5.0 | 3 | 3 | 175.8201159087192 | 0.8678429421563197 |

Table 3: Numerical results on perturbed pendulum.

| Integrator | h | DOFs | R | Maximum Relative $q$ Error | Maximum Relative Hamiltonian Error |
|---|---|---|---|---|---|
| VISE | 1.0 | 3 | 8 | 0.958326009768895 | 0.0036900859853108503 |
| Implicit Midpoint | 1.0 | 2 | 1 | 408.4943461689427 | 0.00018173956693630272 |
| Galerkin Variational Integrator | 1.0 | 8 | 8 | 2.8124033394032035e-11 | 9.994229916836448e-15 |
| | 1.0 | 3 | 3 | 0.8156471866683961 | 1.0207476695695827e-5 |
| VISE | 2.0 | 3 | 16 | 0.6229415651765513 | 0.0002634762006224551 |
| Implicit Midpoint | 2.0 | 2 | 1 | 187.973626902229 | 0.0006500189436792027 |
| Galerkin Variational Integrator | 2.0 | 16 | 16 | 3.139181620903172e-11 | 3.5169568568108006e-14 |
| | 2.0 | 3 | 3 | 5.717753306169191 | 0.00017675442721218666 |
| VISE | 5.0 | 3 | 16 | 0.6891016980719833 | 0.00027585653928133627 |
| Implicit Midpoint | 5.0 | 2 | 1 | 29.434987737731767 | 0.00299934550801042 |
| Galerkin Variational Integrator | 5.0 | 16 | 16 | 2.6562231206550306e-6 | 2.2771663101652665e-15 |
| | 5.0 | 3 | 3 | 107.14217355690201 | 0.01299039004888479 |

Table 4: Numerical results on Hénon-Heiles potential. Note that the degrees of freedom in the table indicate those of each dimension.

| Integrator | h | DOFs | R | Maximum Relative $q$ Error | Maximum Relative Hamiltonian Error |
|---|---|---|---|---|---|
| VISE | 1.0 | 4 | 16 | 4.838439960271346 | 0.8457792975850522 |
| Implicit Midpoint | 1.0 | 2 | 1 | 22.23736998471603 | 0.02454973774617474 |
| Galerkin Variational Integrator | 1.0 | 16 | 16 | 3.2170827801435854e-12 | 1.1600263760322324e-13 |
| | 1.0 | 4 | 4 | 0.02324023997237639 | 2.1471741868377268e-5 |
| | | | | | |
| VISE | 2.0 | 4 | 16 | 13.731349147686684 | 2.210078582450154 |
| Implicit Midpoint | 2.0 | 2 | 1 | 22.256999394783843 | 0.06445840967461561 |
| Galerkin Variational Integrator | 2.0 | 16 | 16 | 3.813551971991108e-8 | 7.050811547518634e-14 |
| | 2.0 | 4 | 4 | 1.214005076679101 | 0.002895264084055545 |
| | | | | | |
| VISE | 5.0 | 4 | 16 | 9.436204346388278 | 1.509118731831439 |
| Implicit Midpoint | 5.0 | 2 | 1 | 445.14675893959543 | 137551.13024621064 |
| Galerkin Variational Integrator | 5.0 | 16 | 16 | 0.0023669760801705284 | 1.99400308093313533e-11 |
| | 5.0 | 4 | 4 | 1.8613623810430068e20 | 5.112395785609402e77 |

# E    COMPARISONS ON ENERGY CONSERVATION WITH INITIAL SYMBOLIC EXPRESSIONS

The initial symbolic expression itself is a parsimonious approximation of the generalized trajectory, but it does not respect the structure of system and thus preserve the energy of the system. The figures below add the energy and generalized trajectory obtained from the initial symbolic expressions and momentum by Legendre transform on top of the previous figures. We could observe that the energy conservation of VISE is better than the initial symbolic expressions.

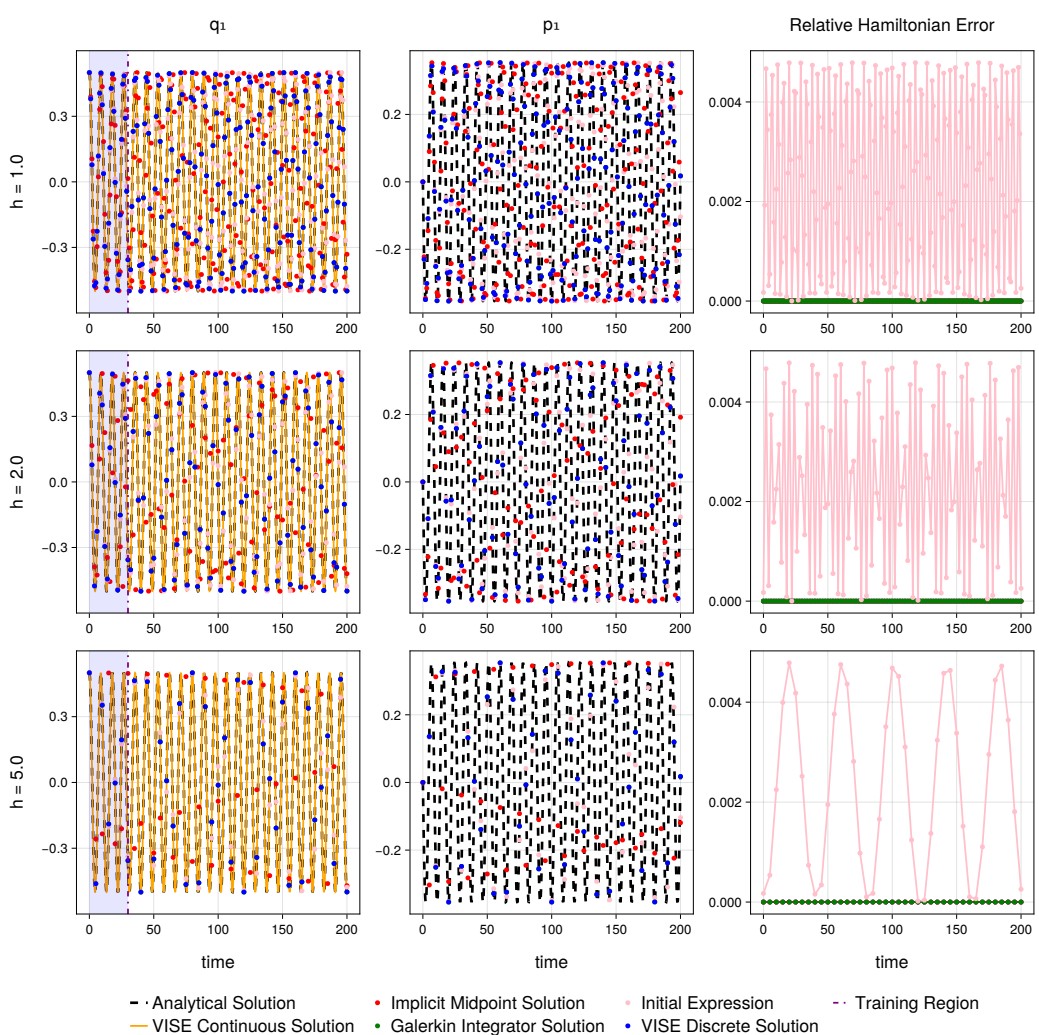

Figure 5: **Harmonic Oscillator.** Though the initial symbolic expression has the same form with analytical solutions, the energy error along the trajectory is bigger than VISE. In other words, by solving the discrete Euler-Lagrange equations, VISE is able to adjust the parameters in the symbolic expression to better preserve the energy of the system.

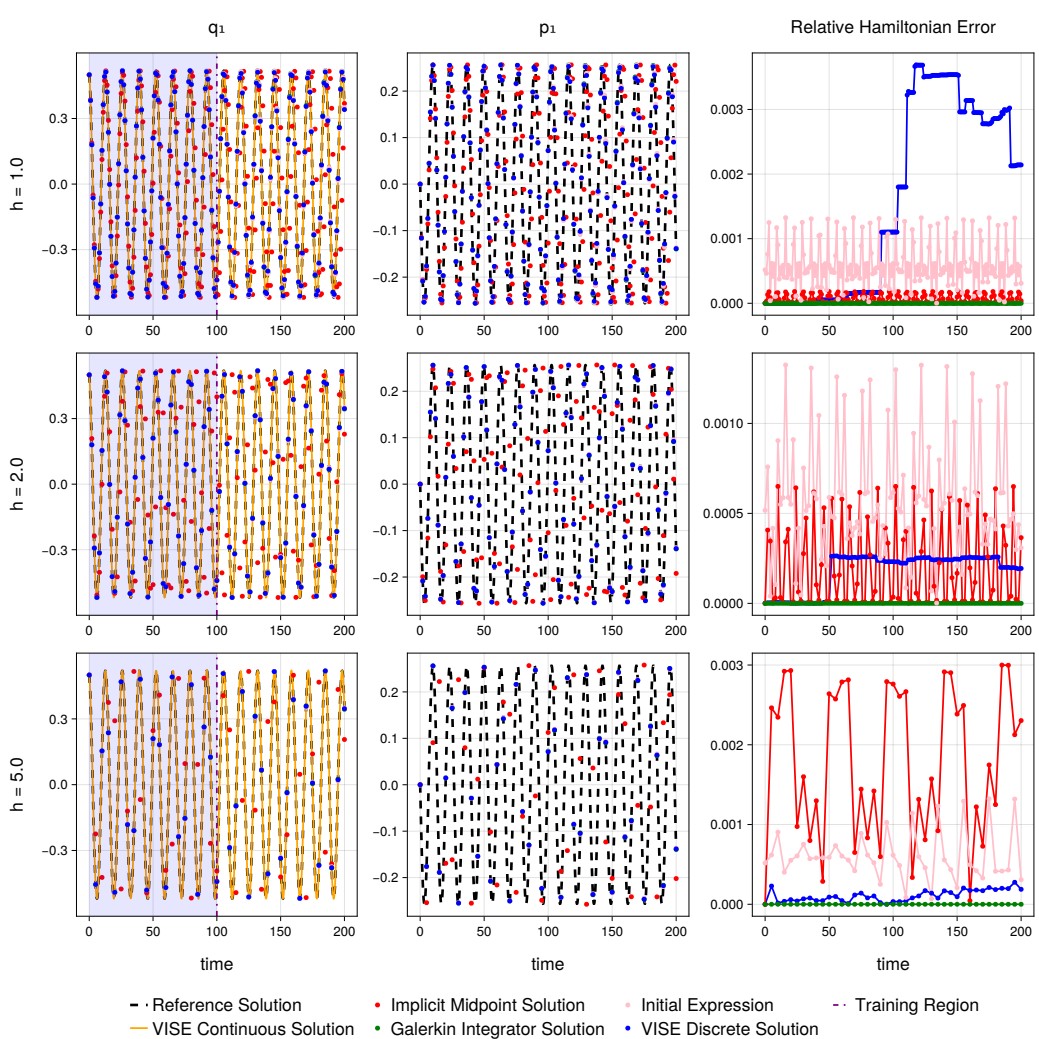

Figure 6: **Perturbed Pendulum.** Similarly, smaller energy error is observed in VISE than the initial symbolic expression for $h = 2.0, 5.0$.

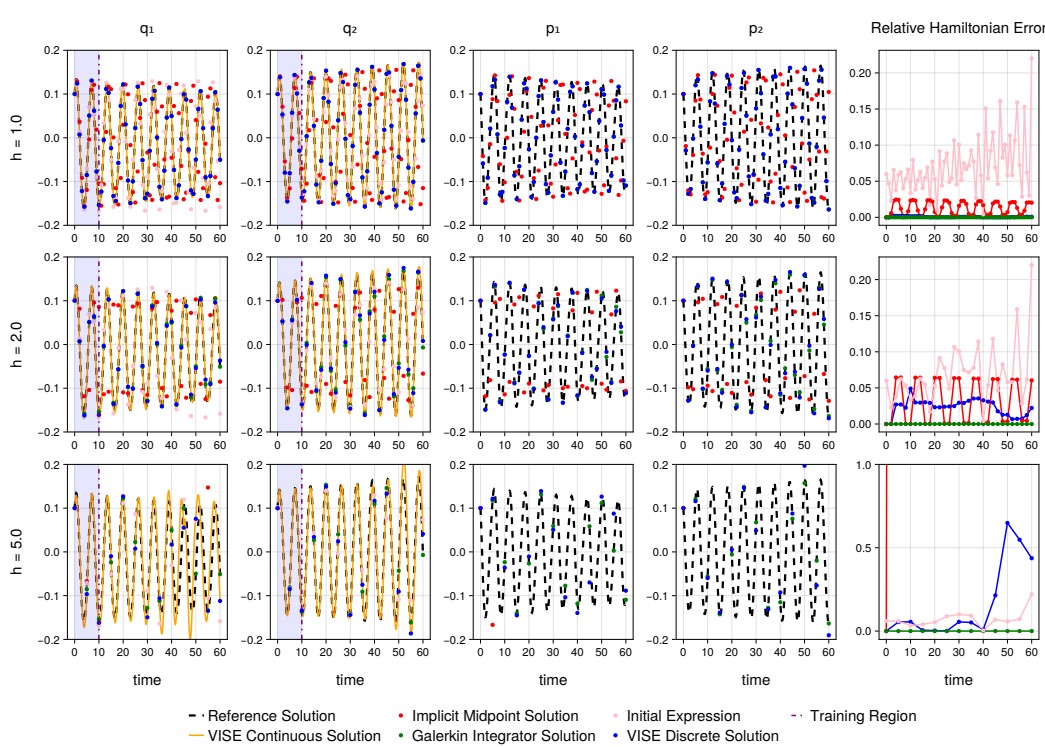

Figure 7: **Hénon-Heiles Potential.** As the initial symbolic expression is only obtained on a narrow time interval, and the amplitude of the generalized trajectory is not constant, the energy error is growing over time for the initial symbolic expression. Instead, VISE is able to adjust the parameters in the symbolic expression to better preserve the energy of the system.

