# OpenReview forum: "VISE:Variational Integration with Symbolic Expressions"
_ICLR.cc/2026/Conference — ICLR 2026 Conference Withdrawn Submission_

### Official Review · Reviewer_GFTU · 2025-10-28

**Soundness:** 2
**Presentation:** 3
**Contribution:** 1
**Rating:** 2
**Confidence:** 4

**Summary:**

The paper presents VISE, a method that replaces the trial functions of a Galerkin Variational Integrator with symbolic expressions $q$ on each interval $[t_n, t_{n+1}]$. The skeleton of these expressions is found using symbolic regression on an interval of a trajectory and then the parameters of the expressions are re-solved at each step using a discrete Euler-Lagrange equations from a quadrature approximation of the action, with the endpoint continuity enforced through Lagrange multipliers. The authors claim that this approach constructs a piecewise continuous and structure preserving integrator. They test VISE on three Hamiltonian systems including a Harmonic Oscillator, Perturbed Pendulum, and Henson-Helios Potential.

**Strengths:**

- The paper presents an interesting interface between symbolic regression and variational integrators. While these are both well established techniques, considering a symbolic expression as adaptive basis that is updated via the Euler-Lagrange equations is novel combination of symbolic AI and geometric integration.

- Symbolic regression approaches discover a global expression with fixed parameters, but VISE treats symbolic expressions as local approximation functions with parameters that evolve with the time intervals. This addresses the limitation of symbolic regression for long time evolution of dynamical systems.

- The mathematical derivations are accessible and well presented except for some minor typos in mathematical notation.

- VISE has the potential to include domain knowledge about system behaviour through the symbolic regression design which might be powerful for specific applications.

**Weaknesses:**

- VISE guarantees symplecticity because it considers a discrete variational principle via the DEL equations. However they do not discuss the conditions under which this holds. This holds assuming that the DEL system is solved to tolerance (requirement 1) and no correction, constraint or other action is performed outside the variational framework after the DEL solve (requirement 2). It is not shown if the first requirement holds.

- Moreover, there is no momentum preservation statement shown and the energy preservation is questionable as in Table 4 of the Henon-Heiles experiments the Hamiltonian error reaches very high values.

- There is no automatic selection of the symbolic expression, as the authors claim that they manually chose a symbolic expression with good accuracy (line 266) which doesn’t support the claim for an automated data-driven process. Also, it is not transparent what that statement means, they might tried a bunch of the discovered functions and picked the one that works.

- For the harmonic oscillator example the initial symbolic expression is essentially the analytical solution form, which means that comparing  with a Galerkin VI is fundamentally unfair, because it encodes domain knowledge. It would me fair to consider a spectral Galerkin with a sin basis function to make it fair.

- On the Henon-Heiles example the VISE method presents very large errors both in trajectories and Hamiltonian while Galerkin VI is shows machine precision accuracy. This results contradict the superiority statement.

- The authors provide no analysis of the overall computational cost of VISE.

- There is no comparison to other methods such as Hamiltonian Neural Networks (Greydanus 2019), Lagrangian Neural Networks (Cranmer 2020), SympNets (Jin 2020) and no consideration of higher dimensional benchmarks such as coupled oscillators. The authors only consider conservative Hamiltonian systems.

- Maybe I am mistaken but VISE is a Galerkin VI with a different basis. I believe that this should be made clearer as it is directly related to the novelty.

- The authors claim that VISE is suitable for high dimensional problems but evidence is not provided to support this.

**Questions:**

- If the authors run VISE with 10 different seeds and choose the three best performing basis functions to evolve and report the errors. You need to quantify if the process is accurate and what is the failure rate when no human is considered to choose the basis.

- What is the wall-clock time of VISE?

- How does VISE compare to Galerkin VI when a trigonometric basis is considered for the harmonic oscillator example?

- If you consider a high dimensional example, e.g. dimensions > 5, how does VISE perform ? It seems to me that the SR has a major bottleneck for coupled and high dimensional problems.

- Why does VISE show that high errors for Henon-Heiles?

- How reliable is the solver? What is the Newton Solver failure rate across all examples? The authors refer to convergence issues of the nonlinear solver but no systematic analysis is performed.  What happens if the solver fail?

- How does VISE compare to the methods listed in the previous section in terms of accuracy and training time?

---

### Official Review · Reviewer_Lity · 2025-10-29

**Soundness:** 2
**Presentation:** 3
**Contribution:** 3
**Rating:** 4
**Confidence:** 4

**Summary:**

This paper introduces VISE (Variational Integration with Symbolic Expressions), a hybrid numerical integration framework that merges symbolic regression with variational integrators for solving Hamiltonian and Lagrangian ODEs. Rather than relying on a single global symbolic model, VISE employs piecewise symbolic expressions whose parameters are dynamically updated at each time step by solving discrete Euler–Lagrange equations. This design preserves symplectic structure, enhances long-term energy conservation, and maintains interpretability. Experimental results on classical systems, including harmonic oscillators, perturbed pendulums, and the Henon–Heiles model, demonstrate that VISE achieves superior accuracy and stability compared to Galerkin variational integrators and implicit midpoint methods, particularly with larger time steps.

**Strengths:**

The paper introduces an original hybrid framework (VISE) that innovatively combines symbolic regression with variational integrators. This design represents a meaningful step beyond both traditional numerical methods and contemporary SciML approaches. By moving away from static symbolic regression models and instead employing piecewise symbolic expressions updated at each time step, VISE preserves symplectic structure and significantly improves long-term energy stability. The authors carefully ground their method in discrete Euler–Lagrange equations, ensuring that symplecticity and conservation properties are respected. This methodological clarity helps distinguish VISE from more heuristic ML approaches. In addition, the symbolic expressions provide transparent functional forms that can offer insight into system behavior.

On benchmark systems such as the harmonic oscillator, perturbed pendulum, and Henon–Heiles potential, VISE consistently achieves excellent accuracy and energy conservation compared to Galerkin variational integrators and implicit midpoint methods, particularly at larger time steps.

**Weaknesses:**

All experiments are conducted on relatively simple, low-dimensional Hamiltonian systems. While these serve as clear proofs of concept, the absence of demonstrations on high-dimensional, stochastic, or PDE-based problems leaves questions about whether the framework can generalize to more challenging real-world applications. Also, the comparisons focus only on classical variational integrators and implicit midpoint methods and comparisons with other SciML approaches (e.g., PINNs, neural operators) are missing

On the methodological side, the pretraining step for symbolic expressions is not sufficiently clarified. The paper defers heavily to external work and manually selects symbolic forms judged to have “good balance” between accuracy and complexity. This manual choice introduces potential selection bias, especially in cases where closed-form solutions are already known (e.g., harmonic oscillator). A more systematic or automated strategy for expression selection would strengthen the approach and its reproducibility.

While mathematically rigorous, the derivations of the discrete Euler–Lagrange updates are dense and may be challenging for a broader ICLR audience. Providing additional intuition, simplifications, or worked examples would make the method more approachable.

Finally, there are some minor issues of polish and precision. For example, line 173 contains a typo (“derivation” instead of “deviation”), and the “Pretraining the Symbolic Expressions” section would benefit from a self-contained explanation rather than relying on external references. These issues do not undermine the contribution, but detract slightly from overall clarity.

**Questions:**

1.	How does computational cost scale with increasing system dimension and symbolic expression complexity?
2.	Can VISE be extended to stochastic systems or settings with noisy observations?
3.	How does VISE compare in performance and interpretability to PINNs or neural operators?
4.	Is the symbolic representation unique? If multiple expressions are valid, how is the selection handled, and could bias arise from manual choices?
5.	Could the authors clarify the symbolic regression process within the paper rather than referring readers externally?

---

### Official Review · Reviewer_BVzf · 2025-10-31

**Soundness:** 2
**Presentation:** 1
**Contribution:** 2
**Rating:** 4
**Confidence:** 2

**Summary:**

This paper studies the problem of efficiently solving Lagrangian and Hamiltonian ordinary differential equations. Compared with variational methods, this paper proposes a hybrid numerical integrator that can maintain interpretability and inherit the symplectic structure of Hamiltonian systems.

**Strengths:**

1. The problem that this paper looks into is interesting and important in a wide range of scientific and engineering domains.

2. The theory in this paper seems to be correct, and the proposed method aligns with the derived equations.

**Weaknesses:**

1. Some related works are not mentioned in this paper, e.g., [1].

2. The experiments in this paper are simple, and the results in this paper are not very observable with only the plot results. Moreover, more baselines should be included, which are mentioned in the introduction part.

3. The writing of this paper is not very good, which makes it hard to look into the challenges in this paper.

[1] Mathiesen, F. B., Yang, B., & Hu, J. (2022). Hyperverlet: A Symplectic Hypersolver for Hamiltonian Systems. Proceedings of the AAAI Conference on Artificial Intelligence, 36(4), 4575-4582.

**Questions:**

1. Can the proposed methods be used in more complex Hamiltonian systems?

2. Can the authors give more numerical results with clear metrics?

---

### Note · Authors · 2025-12-01

I have read and agree with the venue's withdrawal policy on behalf of myself and my co-authors.